# Defective Proinsulin Handling Modulates the MHC I Bound Peptidome and Activates the Inflammasome in β-Cells

**DOI:** 10.3390/biomedicines10040814

**Published:** 2022-03-30

**Authors:** Muhammad Saad Khilji, Pouya Faridi, Erika Pinheiro-Machado, Carolin Hoefner, Tina Dahlby, Ritchlynn Aranha, Søren Buus, Morten Nielsen, Justyna Klusek, Thomas Mandrup-Poulsen, Kirti Pandey, Anthony W. Purcell, Michal T. Marzec

**Affiliations:** 1Department of Biomedical Sciences, University of Copenhagen, 2200 Copenhagen, Denmark; saad.khilji@sund.ku.dk (M.S.K.); carolin.hoefner@sund.ku.dk (C.H.); tmpo@sund.ku.dk (T.M.-P.); 2Department of Biochemistry and Molecular Biology, Biomedicine Discovery Institute, Monash University, Clayton, VIC 3168, Australia; ritchlynn.aranha1@monash.edu (R.A.); kirti.pandey1@monash.edu (K.P.); 3Department of Physiology, University of Veterinary and Animal Sciences, Lahore 54000, Pakistan; 4Department of Medicine, School of Clinical Sciences, Monash Univesity, Clayton, VIC 3168, Australia; pouya.faridi@monash.edu; 5Department of Pathology and Medical Biology, University Medical Center Groningen, University of Groningen, 9713 Groningen, The Netherlands; e.pinheiro.machado@umcg.nl; 6Laboratory of Translational Nutrition Biology, Department of Health Sciences and Technology, Institute of Food, Nutrition and Health, ETH Zürich, 8603 Zürich, Switzerland; tina.dahlby@hest.ethz.ch; 7Department of Immunology and Microbiology, University of Copenhagen, 2200 Copenhagen, Denmark; sbuus@sund.ku.dk; 8Department of Health Technology, Section for Bioinformatics, Technical University of Denmark, 2800 Lyngby, Denmark; morni@dtu.dk; 9Instituto de Investigaciones Biotecnológicas, Universidad Nacional de San Martín, San Martín CP1650, Argentina; 10Laboratory of Medical Genetics, Department of Surgical Medicine, Collegium Medicum, Jan Kochanowski University, 25-369 Kielce, Poland; jsklusek@ujk.edu.pl; 11Institute of Health Sciences, Collegium Medicum, Jan Kochanowski University, 25-002 Kielce, Poland

**Keywords:** ER stress, GRP94, proteasome, insulin, inflammation, MHC class-I

## Abstract

How immune tolerance is lost to pancreatic β-cell peptides triggering autoimmune type 1 diabetes is enigmatic. We have shown that loss of the proinsulin chaperone glucose-regulated protein (GRP) 94 from the endoplasmic reticulum (ER) leads to mishandling of proinsulin, ER stress, and activation of the immunoproteasome. We hypothesize that inadequate ER proinsulin folding capacity relative to biosynthetic need may lead to an altered β-cell major histocompatibility complex (MHC) class-I bound peptidome and inflammasome activation, sensitizing β-cells to immune attack. We used INS-1E cells with or without GRP94 knockout (KO), or in the presence or absence of GRP94 inhibitor PU-WS13 (GRP94i, 20 µM), or exposed to proinflammatory cytokines interleukin (IL)-1β or interferon gamma (IFNγ) (15 pg/mL and 10 ng/mL, respectively) for 24 h. RT1.A (rat MHC I) expression was evaluated using flow cytometry. The total RT1.A-bound peptidome analysis was performed on cell lysates fractionated by reverse-phase high-performance liquid chromatography (RP-HPLC), followed by liquid chromatography coupled with tandem mass spectrometry (LC-MS/MS). The nucleotide-binding oligomerization domain, leucine rich repeat and pyrin domain containing protein (NLRP1), nuclear factor of kappa light polypeptide gene enhancer in B-cells inhibitor alpha (IκBα), and (pro) IL-1β expression and secretion were investigated by Western blotting. GRP94 KO increased RT1.A expression in β-cells, as did cytokine exposure compared to relevant controls. Immunopeptidome analysis showed increased RT1.A-bound peptide repertoire in GRP94 KO/i cells as well as in the cells exposed to cytokines. The GRP94 KO/cytokine exposure groups showed partial overlap in their peptide repertoire. Notably, proinsulin-derived peptide diversity increased among the total RT1.A peptidome in GRP94 KO/i along with cytokines exposure. NLRP1 expression was upregulated in GRP94 deficient cells along with decreased IκBα content while proIL-1β cellular levels declined, coupled with increased secretion of mature IL-1β. Our results suggest that limiting β-cell proinsulin chaperoning enhances RT1.A expression alters the MHC-I peptidome including proinsulin peptides and activates inflammatory pathways, suggesting that stress associated with impeding proinsulin handling may sensitize β-cells to immune-attack.

## 1. Introduction

Type 1 diabetes (T1D) is characterized by the autoimmune destruction of β-cells as a consequence of a break in tolerance towards β-cell self-peptides. How this break in tolerance is attained still remains to be answered. Recent evidence points to the emergence of neoepitopes as a consequence of β-cell stress [1,2]. As professional secretory cells, β-cells are challenged with physiological ER stress during the production and secretion of insulin [3]. In addition to inherent physiological stress, β-cells can be challenged further with environmental factors associated with the onset of T1D, including but not restricted to viral infection, diet, cow’s milk ingestion in early life, and gut microbiota composition [4,5,6,7]. All these factors lack uniformity and loss of tolerance could be related to β-cell secretory stress. Proinsulin (PI) is the major β-cell-translated product as well as a major autoantigen in human and rodent models of type 1 diabetes [8,9]. Recent advances in protein analytical technologies have expanded the list of neoantigens in type 1 diabetes, especially those derived from (pro) insulin. These include normal insulin peptides, hybrid insulin peptides [1,10,11], as well as post-translational modifications (PTM) of insulin chains [12]. This would suggest that nonconventional processing and degradation of insulin may contribute to the progression of T1D or even as a trigger for the loss of the peripheral immune tolerance. During its processing from pre-prohormone to mature insulin, PI requires the assistance of enzymes and chaperones in the ER such as protein disulfide isomerases (PDIs) and glucose-regulated protein (GRP)-94 and -78 to facilitate disulfide bond formation and to attain its native state [13,14,15,16]. Diminished activity of ER chaperones such as GRP94 results in proinsulin mishandling, provokes compensated ER stress in the form of PERK activation [13], and activates inducible proteasomes [17], potentially setting the stage for inflammasome activation and auto-epitope formation and presentation. Studies in the non-obese diabetic (NOD) mouse have shown increased β-cell stress before the onset of clinical diabetes [18]. Similarly, human leukocyte antigen (HLA) hyperexpression and the presence of insulitis has been observed in human islets from recent-onset T1D patients [19], suggesting β-cell stress precedes inflammatory events in the pancreas.

The incidence of T1D is increasing among children and adolescents [20]. Although genetics account for roughly 50% of total T1D development risk, its recent increased incidence suggests an increased influence of environmental factors. A popular hypothesis supports a role of β-cells dysfunction at multiple levels in their own demise [21,22,23,24,25,26,27]. Among these defects are believed to be those that affect (pro) insulin-folding resources and induce ER stress (e.g., growth, puberty, pregnancy, insulin resistance) [28]. It is known that ER stress can induce secretion of the ER-resident heat shock protein (HSP) family of chaperones that can be taken up by antigen-presenting cells (APC) [29,30], allowing cross-presentation and activation of CD8^+^ T-cells [31]. Cross presentation thereby bypasses the conventional APC-Th cell-dependent immune activation and sensitizes target cells exposing auto-epitope on MHC I to T-cell attack. GRP94 (an HSP90 family member) has been found, free- or IgG-bound, circulating in people with T1D [32] and can efficiently cross-present chaperoned peptides to CD8^+^ T-cells [33]. This suggests that activation of ER stress in response to environmental factors, followed by (or leading to) secretion of GRP94-PI complexes, may trigger the β-cell-directed immune response.

Recently, we discovered that the β5i subunit of the intermediate proteasome is over-expressed in GRP94 deficient β-cells and its inhibition restores proinsulin levels [17]. Employing GRP94 KO/i and cytokines IL-1β/IFNγ exposure to mimic ER or inflammatory stress, we found these conditions increased RT1.A expression in β-cells. Correspondingly, the total RT1.A bound peptide repertoire increased upon these treatments and included the presentation of peptides derived from (pro) insulin. GRP94 KO also increased NLRP1 inflammasome expression, reduced cellular IκBα levels and led to a reduction in pro-IL-1β protein content coupled with an increase in mature IL-1β secretion, indicating activation of a β-cell inflammatory response.

## 2. Materials and Methods

### 2.1. Procurement of Cell Lines and Reagents

The mouse hybridoma cell line OX-18 for generating anti-RT1.A (anti-rat MHC I) antibody was purchased from CellBank, Australia. The rat insulinoma cell line (INS-1E) was a kind gift from Claes Wollheim (University Medical Center, Geneva, Switzerland). Protein G resin was purchased from Agarose Bead Biotechnologies (Cat# 4RRPG-100, Madrid, Spain). All other chemicals/reagents of HPLC/MS grade were purchased from commercial vendors and are listed in Appendix A.

### 2.2. Cell Culture and Pelleting

The unchallenged INS-1E and their derived GRP94 KO and control KO (cells targeted with lentiviral particles still expressing GRP94 similar to unchallenged INS-1E) cells were cultured and maintained between passage numbers 54–72 in RPMI-1640 as described previously [17]. To perform immunopeptidomics, cells from confluent flasks were detached using EDTA (10 mM), centrifuged at 4000 rpm to pellet the cells. The cell pellets of 5 × 10^8^ per replicate per group were snap-frozen in liquid nitrogen and stored at −80 °C until later use for immune-affinity capture of RT1.A.

The hybridoma cell line OX-18 was cultured in roller bottles in RF5 medium (RPMI-1640 supplemented with 5% fetal bovine serum (FBS), 1% Penicillin-Streptomycin, 2 mM MEM non-essential amino acids, 100 mM HEPES, 2 mM L-glutamine and 50 µM β-mercaptoethanol). The supernatants were collected and antibody harvested by protein G chromatography using a Profinia purification system (Bio-rad^®^, South Granville, Australia).

### 2.3. Drugs Treatment/Cytokines Exposure

To induce GRP94 functional loss or mimic inflammatory conditions, INS-1E cells were treated with either 20 µM GRP94 inhibitor (GRP94i) PU-WS13 (EMD Millipore Corp.^®^, Burlington, MI, USA) or exposed to recombinant human interleukin-1β (IL-1β, Sino Biological Inc.^®^, Beijing, China) at 15 pg/mL or 10 ng/mL recombinant rat IFNγ (R&D^®^, Minneapolis, MN, USA) for 24 h prior to harvesting cells for flow cytometry and collection of pellets for immunopeptidomics.

### 2.4. Flow Cytometry

The unchallenged or GRP94 KO/control KO INS-1E cells cultured in the absence or presence of drug/cytokines at the dose and time mentioned above were washed with FACS buffer (2% FBS in PBS) and incubated with OX-18 for 1 h at 4 °C. The cells were again washed with FACS buffer followed by incubation with phycoerythrin (PE)-labeled anti-mouse antibody for 30 min. The cells were then re-suspended in 300 µL FACS buffer for flow cytometry. The flow cytometry data for RT1.A was acquired by LSRII (BD Biosciences^®^, Franklin Lakes, NJ, USA). The data were analyzed using flowjo^®^ software (Tree Star Inc.^®^, Ashland, OR, USA). The cells were gated on SSC-A × FSC-A and FSC-H × FSC-A to exclude doublets and debris, and the mean fluorescence intensity (MFI) for each group was recorded at 585 nm.

### 2.5. Isolation of RT1.A Bound Complexes

The detailed procedures for antibody cross-linking and purification of MHC bound complexes and fractionation have been described in detail previously [34]. Briefly, the frozen INS-1E cell pellets were ground using a cryogenic mill (Retsch Mixer Mill MM 400) and lysed in buffer containing 0.5% IGEPAL (Sigma-Aldrich^®^, St. Louis, MO, USA), 50 mM Tris pH 8, 150 mM NaCl and supplemented with protease inhibitor cocktail (Roche^®^, Germany). The lysate was ultra-centrifuged (Beckman Coulter Optima L90K 77LR271) at 40,000× *g*, 4 °C for 2 h. The supernatant was collected, and RT1.A complexes were immunoaffinity purified by passing through a column containing cross-linked OX-18 antibody coupled to protein G agarose resin (ABT Beads^®^, Madrid, Spain). The bound complexes were washed with buffers with decreasing detergent and salt concentrations as detailed [34]. The MHC-bound peptides were eluted in 10% acetic acid for fractionation.

### 2.6. Fractionation by Reverse-Phase High-Performance Liquid Chromatography (RP-HPLC)

The peptide eluents were fractionated by RP-HPLC using a C18 column (Chromolith Speed Rod, Merck-Millipore^®^, Burlington, MA, USA) and an ÄKTA micro HPLC system (GE Healthcare^®^, Chicago, IL, USA) running a mobile phase consisting of buffer A (0.1% trifluoroacetic acid (TFA)) and buffer B (80% acetonitrile/0.1% TFA). The flowrate was adjusted at 2 mL/min to collect peptides in 1 mL fractions in low-protein binding Eppendorf tubes (Eppendorf LoBind^®^ tubes, cat. no. EPPE0030108.116, Hamburg, Germany). Peptide-containing fractions were collected, vacuum concentrated on speedVac (Labconco^®^, Kansas City, MO, USA) to ~5 µL and combined using a fraction concatenation strategy into 5 pools per replicate. The pools were reconstituted in 0.1% formic acid +2% acetonitrile (final volume = 10 µL) and stored in −80 °C until LC-MS/MS analysis.

### 2.7. Mass Spectrometry Data Acquisition and Analysis

The pooled samples were sonicated, centrifuged at 15,000 rpm for 30 min and transferred to mass spectrometry vials, each pool spiked with indexed retention time (iRT) peptides to normalize retention time. The samples were run on a Orbitrap fusion Tribrid LC-MS/MS (Thermo^®^, Waltham, MA, USA) using data-dependent acquisition (DDA) strategy as previously described [35].

The .raw files acquired from the system were analyzed using PEAKS (PEAKS X, Bioinformatics solutions Inc.^®^, Waterloo, ON, Canada) and spectra searched against the rat proteome (Uniprot 8133 reviewed entries by 30 November 2021). The parameters for peptide spectral matching included error tolerance of 10 ppm and 0.02 Da tolerance for fragment ions, enzyme was set to none, unspecific digestion mode, and the PTMs included variable methylation, cysteinylation, and deamidation (NQ). The list of identified peptides and proteins were exported as .csv files at 5% FDR.

### 2.8. Bioinformatics Analyses

The consensus RT1.A binding motifs were created using IceLogo [36], Venn diagrams by Bio Venn [37] and unsupervised clustering for alignment of RT1.A-bound 8–15 mer peptide sequences using Gibbs Clustering [38]. The consensus binding motif is presented for two out of five Gibbs Clusters due to limited peptide yield in some groups. For GRP94 KO analysis, two independent clones each run in duplicate were combined, keeping the peptides common between the two clones to avoid clonal variation.

To construct an artificial neural networks-based model to predict peptide-MHC interaction independent of MHC, the tool NNAlign [39] was trained on a combination of random natural peptides (assigned value = 0) and INS-1E peptides from our experiments (excluding proinsulin sequences, assigned value = 1) with maximum overlap for common motif changed to 8 amino acids. The proinsulin sequences from experimental data mixed with random natural peptides were then run against the trained model and compared to get percentile rank values against the trained model.

### 2.9. Tryptic Digestion of Heavy Chain

To confirm whether INS-1E cell line expresses both class Ia and class Ib alleles, we performed tryptic digestion of the heavy chain fragment obtained from RP-HPLC fractionation. Briefly, the heavy chain fragments were pooled, vacuum-dried to reduce volume, and digested overnight with 2 µg trypsin in suspension trap (S-trap) columns (Profiti^®^, Durres, Albania) according to manufacturer’s instructions. For peptide recovery, 50 mM tetraethyl ammonium bicarbonate buffer was added to the digest followed by 0.2% formic acid and finally 50% acetonitrile with 0.2% formic acid. The sample was centrifuged between each step. The total sample volume was reduced to ~5 µL in a vacuum centrifuge and reconstituted in 0.1% formic, 2% acetonitrile solution (final volume = 10 µL) to run on LC-MS/MS (Triple-TOF 6600 system, SCIEX^®^, Framingham, MA, USA). For proteomics analysis of the tryptic digest of heavy chains against reviewed rat proteome, the settings on PEAKS X^®^ included MS1 tolerance 15 ppm MS2 tolerance 0.1 Da and PTMs included: fixed carbamidomethylation of cysteine variable: Methionine (Oxidation), N-terminal acetylation. Peptide identity data was exported for analysis at 5% FDR cutoff as .csv files.

### 2.10. Western Blotting

The Western blotting was performed following the procedure detailed in [40]. In brief, 4 million of the confirmed CRISPR/Cas-9 targeted GRP94 KO INS-1E and control KO cells (CRISPR/Cas-9 targeted INS-1E cells still expressing GRP94 similar to unchallenged INS-1E) were grown for 48 h in T25 flasks. Before lysing cells, the supernatant was collected to estimate secreted IL-1β. The cells were then lysed in buffer supplemented with a protease inhibitor cocktail (Life Technologies, Nærum, Denmark). The protein concentration was calculated using Bradford assay (Bio-Rad, Copenhagen, Denmark). Thirty µg protein was loaded for each sample on 4–12% bis-tris gel and transferred to PVDF membrane (iBLOT2 system^®^). The membranes were blocked in 5% milk in TBST buffer and incubated overnight with antibodies against proinsulin, tubulin, NLRP1, IκBα, IL-1β and GRP94 as listed in Appendix A along with the dilution used, which was followed by 1 h incubation with anti-rat, anti-mouse, or anti-rabbit antibodies depending on the primary antibody host. The membranes were developed after washing 3× in TBST. The chemiluminescent images were acquired using Azure^®^ Saphire Biomolecular Imager. For supernatant Western blots to estimate secreted IL-1β, the procedure was performed as described previously by [13]. Briefly, 500 µL of media was centrifuged at 14,000 rpm for 30 min using 10 kDa filter to concentrate IL-1β on filter in ~40 µL volume. The supernatants were then mixed with lysis buffer and loading buffer and run on gel as described above. The images were quantified for protein expression using ImageJ (v 1.52a) and the ratio for the proteins of interest to tubulin was used for statistical comparison. For IL-1β quantification from supernatants, only acquired bands were quantified and compared between the groups.

### 2.11. Statistical Analysis

The flow cytometry data were analyzed using paired *t*-test for treatment versus relevant control groups using statistical software GraphPad Prism version 9.1.0 (La Jolla). For Western blots, an unpaired *t*-test was used on quantified blots for statistical comparison. The number of * in figures refer to the level of significance of results between the groups where *, ** and *** reflect *p* ≤ 0.05, *p* ≤ 0.01, and *p* ≤ 0.001, respectively.

### 2.12. Data Availability

The data acquired by mass spectrometry will be deposited at ProteomeXchange Consortium via the PRIDE data repository [41]. A request for raw data and search results can also be sent to anthony.purcell@monash.edu.

## 3. Results

### 3.1. INS-1E Cells Predominantly Express Class Ia Allele

Rat RT1 system has two MHC I regions. RT1.A contains the centromeric, classical Ia region whereas the telomeric, nonclassical Ib region is referred as RT1.C/E/M (Appendix A). OX-18 antibody is specific for RT1.A but also known to cross-react with rat non-classical Ib products [42], though class Ib products are generally poorly expressed in different tissues [43]. We performed tryptic digestion of the heavy chain fragments from the fractionated RT1 immunoprecipitate to explore which class I regions were captured by the OX-18 antibody from INS-1E cells. We found classical class Ia derived peptides in the tryptic digest, suggesting it is the dominantly expressed class I allele in INS-1E cells (Appendix A).

### 3.2. GRP94 KO/Cytokines Exposure Increase RT1.A Expression

To estimate if GRP94 KO/cytokine exposure altered overall RT1.A expression, INS-1E cells were cultured with/without the cytokines IL-1β/IFNγ or in the presence of GRP94i or GRP94 KO. The cells under tested conditions were incubated with OX-18 antibody followed by incubation with PE-labelled secondary antibody. Unchallenged INS-1E cells showed considerable basal RT1.A expression with mean fluorescence intensity (MFI) around 1000. As expected, exposure to cytokine IL-1β or IFNγ significantly upregulated RT1.A expression; likewise, GRP94i also modestly upregulated RT1.A expression (Figure 1A,C). Interestingly, GRP94 KO significantly increased RT1.A expression (Figure 1B,D) while GRP94i treatment of INS-1E cells showed a modest upregulation in MFI (*p*-value = 0.08).

### 3.3. GRP94 KO Modulates RT1.A-Bound Peptides Yield

The unchallenged INS-1E cells or GRP94i-treated/cytokine-exposed/GRP94 KO INS-1E cells were lysed for immunoaffinity capture of RT1.A. The RT1.A bound peptides were fractionated and then subject to LC-MS/MS. The immunopeptidomics was carried out on two biological replicates for each group. Taking all groups together, we found 4505 peptides (derived from 1653 source proteins, as listed in Appendix A) that were overlapping or unique to different tested conditions.

Some rat strains, for example AVN-carrying RT1.A^a^ haplotype, can present peptides of 8–15 amino acids in length [44]. As no previous information was available for INS-1E cells derived from NEDH rats carrying RT1.A^g^ haplotype [45], we included these same peptide lengths in our analysis. The majority of peptides (83–90%) in each group ranged between 8–15 amino acids in length. The number of peptides obtained ranged from 883 from INS-1E (from 484 source proteins) to 3365 peptides (from 1434 source proteins) in IFNγ-exposed INS-1E cells, whilst the GRP94 KO group showed an intermediate yield of 1652 peptides (from 862 source proteins) (Figure 2A). Out of 1652 peptides found in the GRP94 KO groups, 1063 peptides (from 659 source proteins) were overlapping between the two GRP94 KO clones, confirming low clonal variation. Of note, the peptide overlap between biological replicates for each tested group ranged between 50–75% (Figure 2B).

Peptides present in unchallenged INS-1E and control KO groups were subtracted from GRP94 KO, IL-1β or IFNγ-exposed groups (Figure 3A–C) to identify 362 peptides (from 206 source proteins) unique to GPR94 KO INS-1E cells. Similarly, IL-1β- or IFNγ-exposed INS-1E had 991 and 2416 unique peptides, respectively (Figure 3A–C). We then compared GRP94 KO and cytokine-exposed groups to obtain GRP94 KO and inflammatory setting-exclusive peptides. Fifty-three of these peptides were exclusive to GRP94 KO, while 346 and 1726 peptides were unique to IL-1β- or IFNγ-exposed groups, respectively (Figure 3D). The overlap between GRP94 and cytokine-exposed group peptides suggests some similarity between limited folding capacity and the inflammatory response. We also found peptides from other proteins localized in the secretory granules common between GRP94 KO and cytokines groups, e.g., of islet amyloid polypeptide (IAPP) and chromogranin-A (CMGA). Altogether, these data indicate the ER stress and inflammatory stress partly overlap in their activity to modulate MHC-I peptidome.

### 3.4. RT1.A^g^ Has a Preference for Nonamer Ligands

The length distribution of peptides identified from the OX18 immunopreciptates showed nonamers to comprise the highest proportion (around 55%) in all groups, followed by decamers and undecamers as the second and third most preferred length of peptides for RT1.A^g^ binding (Figure 4).

The consensus RT1.A-binding motifs showed strong preferences for leucine, isoleucine and valine at P2 while arginine and lysine dominated PΩ (Figure 5 and Figure 6). No obvious differences were observed in amino acid preferences within the peptides in any group. The only exception was seen in IFNγ-exposed INS-1E cells where tyrosine appeared as the third most abundant amino acid at PΩ in nonamers (Figure 5 and Figure 6).

The Gibbs cluster analysis for peptides from all tested groups also confirmed consistent binding motifs similar to Icelogo (Appendix A).

### 3.5. GRP94 KO or Cytokine Exposure Modulates Proinsulin Derived RT1.A-Bound Peptides

In addition to the diversification of the peptide repertoire in GRP94 KO and cytokine-exposed INS-1E cells, we also found a qualitative increase in proinsulin-derived peptides. Three peptides derived from proinsulin were found in one or both cytokine-treated groups (VLWEPKPAQAFVK, ILWEPKPAQAFVK, ALWMRFLPL) (Table 1). Moreover, we also found four peptides (HLVEALYL, ALYLVCGERGFF, ALYLVCGERGFFYTP, and YLVCGERGFF) common to GRP94 KO/i and cytokine-exposed groups. The overlap between cytokines and GRP94 KO/i groups suggests a possible interplay between defective folding and inflammatory pathways. Interestingly, we found two (pro) insulin peptides FFYTPKS and VEDPQVPQ unique to GRP94 KO/i groups. The peptide FFYTPKS is a non-canonical 7-mer but was detected in both GRP94 KO/i conditions. The two peptides do not fit the canonical binding motif and can be fragmented versions of a larger peptide but are found exclusively and in the majority of biological replicates of GRP94 KO clones/GRP94i treatment.

We trained the artificial neural networking platform NNAlign to scan our experimental (pro) insulin sequences for the occurrence of the RT1.A^g^ binding motif and to screen them as potential high-scoring binder among INS-1E-derived peptides. The computed binding motif from the NNAlign analysis model was similar to that obtained from Gibbs cluster analysis or Icelogo (Appendix A). In general, we found increased percentile rank values for GRP94 KO/i/cytokines groups (Appendix A). The best computed score as binder in parental INS-1E and control KO clone were ~15 for the peptides LALEVARQ and ALYLVCGERGF, respectively. GFFYPTMS (B:23–30) was the highest-scoring peptide in IFNγ group with a score of 0.741. Interestingly, the B-chain peptides common to GRP94 KO/i/cytokines groups were among the top-scoring peptides despite the fact that they do not conform to the canonical binding motif (Figure 5 and Figure 6). The B-chain peptide HLVEALYL (B:10–B17) appeared as a top-scoring peptide in GRP94 KO/IL-1β groups and second-highest in IFNγ exposed group with a score of ~8. The peptide ALYLVCGERGFF unique to GRP94 KO/i/IFNγ groups appeared as the second-highest-scoring peptide with a percentile rank of 13. Interestingly, GFFYTPKS is the top-scoring peptide (percentile rank = 4.7) in GRP94 inhibitor-treated group. Collectively, these observations indicate an increased likelihood of proinsulin peptides from treatment groups as RT1 binders upon GRP94 KO/cytokines exposure.

### 3.6. Limiting β-Cell Folding Capacity Activates Inflammatory Pathways

To test whether a defect in proinsulin folding capacity in β-cells leads to activation of inflammatory pathways, we tested the expression of the NLRP1, IκBα, and IL-1β content and secretion in our β-cell model. We found that NLRP1 expression (at 165 kDa fully organized NLRP1 and 70 kDa corresponding to NLRP1 lacking leucine-rich repeats (LRR)) was upregulated in GRP94 KO cells compared to the control groups (Figure 7A,C). Typically, NLRP1 consists of leucine-rich repeats (LRR), a NACHT domain, and a PYD/CARD domain. In general, LRR is considered to play a role as the ligand recognition domain for inflammasomes; however, a recent study showed that this domain is not required for NLRP3 activity [46]. The IκBα on the other hand was found to be decreased in the GRP94 KO group (Figure 7D). We also determined cell contents of proIL-1β, a canonical substrate of NLRP1 processing. In accordance with the increased NLRP1 expression, we found that pro-IL-1β content was reduced in GRP94 KO compared to control KO while intracellular mature IL-1β content was hardly detectable but found to be increased as a secreted product. Taken together, these data suggest that β-cell folding incapacity suggest NFκB activation and increased NLRP1 expression and activity leading to proIL-1β processing and secretion of IL-1β.

## 4. Discussion

The long-standing knowledge gap in understanding T1D pathophysiology is how tolerance is lost against β-cell self-peptides, resulting in autoimmune destruction of β-cells. In this study, we provide the proof-of-principle that ER stress as a result of diminished or functional-loss of β-cell folding capacity leads to an increased expression of the MHC-I antigen presentation, modulation of the MHC-I bound peptidome, as well as activation of inflammatory pathways. We also established the first database for haplotype RT1.A^g^ bound peptides that are endogenously processed and presented in a commonly used rat β-cell line, INS-1E.

The ER has a key role in the MHC class I antigen presentation pathway as peptides generated by the proteasome are transported into the ER and loaded onto MHC class I molecules for immune surveillance [47]. Protein misfolding in the ER, e.g., tyrosinase misfolding, has been shown to increase MHC antigen presentation efficiency tyrosinase-derived epitopes [48]. MHC I hyperexpression is a hallmark of T1D as seen in islets from newly diagnosed patients [19,49] and an elevated MHC I expression facilitates antigen presentation to CD8^+^ T-cells [50]. We found an increased RT1.A expression on GRP94 KO β-cells as well as following cytokine exposure (Figure 1A–D). The increased RT1 expression may be mediated via the NF-κB pathway, which itself is activated by ER stress response or cytokines [51,52]. Indeed, GRP94 KO INS-1E cells exhibit PERK/eIF2α activation [13] which may attenuate IκBα translation [53]. This could lead to the activation of NF-κB to promote transcription of proinflammatory genes and upregulate MHC-I expression [54]. The increased MHC-I expression in the islet microenvironment due to ongoing inflammatory processes [55] makes β-cells more visible to the effector T-cells and possibly explains β-cell vulnerability in T1D patients despite the similar frequency of circulating β-cell reactive CD8^+^ T-cells in both T1D patients and healthy individuals [2,56].

We found that GRP94 KO/i diversified the RT1-bound peptide repertoire as did cytokine exposure (Figure 2A,B). The GRP94 KO group peptide repertoire overlapped partly with those from cytokine-exposed groups (Figure 3A–D) suggesting some shared signaling pathways between these two separate inducers of ER stress. Interestingly, the GRP94 KO also shared (pro) insulin peptides with the cytokines group. (Pro) insulin-derived peptides, especially those derived from proinsulin B-chain, are recognized as epitopes and listed, e.g., in [57]. We have found four peptides (HLVEALYL, ALYLVERGERGFF, ALYLVCGERGFFYTP, and YLVCGERGFF) from insulin B-chain (Table 1) common in GRP94 KO and cytokine-exposed experimental groups. Diminished GRP94 activity is expected to affect proinsulin processing [13], as do the cytokines such as IL-1β [58], suggesting ER stress induced by limited folding capacity mimics the action of cytokines on (pro) insulin processing. The two GRP94 KO/i exclusive insulin peptides are VEDPQVPQ (C-peptide: 2–9) and FFYTPKS (B: 25–31) (Table 1). As a defect in GPR94 activity leads to mishandling at the proinsulin level, it is plausible that the C-peptide also contributed to MHC-I peptidome in the GRP94 KO group. The FFYTPKS is a non-canonical peptide but appeared in the majority of mass spec analyses but despite proper setup, it may still be a non-specific binder or a fragment of a larger peptide. The presence of other β-cell proteins such as IAPP (ILVALGHLR) and CMGA (RMQLAKELT) overlapping in both GPR94 KO and cytokine-exposed groups (Appendix A) suggests that the GRP94 functional deficiency impact on protein folding in ER might not be restricted to proinsulin and warrants further investigations.

Studies show ER stress precedes inflammation in pancreas before the clinical onset of T1D [18,59]. The interesting question is how in our experimental settings, limited folding capacity coupled with compensated UPR activation [13] activates inflammation pathways. As our experiments were conducted in vitro, the cells must be reliant on internal mechanisms to activate inflammatory pathways. As PERK has been reported to increase inflammasome expression [60], we tested expression levels of NLRP1 and its downstream targets, IL-1β and IκBα, in our β-cell model. Interestingly, we found an increased expression of NLRP1 in the GRP94 KO group compared to the control group (Figure 7). Typically, inflammasome activation requires two signals. Signal-1 such as glucose, lipopolysaccharides, or cytokines [61,62,63] binding to its receptor is the priming signal that upregulates NLRP subunits and proIL-1β transcription. Signal-2 can be provided by extracellular ATP, toxins, and particulate matter such as IAPP [64,65,66], which subsequently activates the inflammasome. This leads to activation of caspase-1, which in turn cleaves proIL-1β and proIL-18 into mature and biologically active IL-1β and IL-18. Some drugs that induce ER stress such as thapsigargin and tunicamycin or mitochondrial oxidative stress inducers like rotenone can provide both signal-1 and signal-2 via promoting interaction of NLRP with thioredoxin-interacting protein (TXNIP) [67,68,69,70]. PERK activation in GRP94 KO cells can lead to increased inflammasome activation [60] via TXNIP as in [67]. To confirm NLRP1 overexpression is followed by inflammasome activation, we estimated levels of (pro) IL-1β content as well as secretion. We found decreased pro-IL-1β content in GRP94 KO cells possibly due to secretion as mature IL-1β. Indeed, we found a high amount of IL-1β in the supernatants of GRP94 KO clonal cells (Figure 7B,F), confirming high activity of NLRP1 inflammasome in GRP94 deficient β-cells, possibly facilitating the onset of inflammatory events in the islet microenvironment.

The importance of GRP94 in β-cells has only recently gained attention. GRP94 has long been known as critical for prenatal development and ER quality control [71], as is the case with the pancreas [72]. Conditional knockout of GRP94 in mice leads to pancreatic hypoplasia and reduced β-cell mass and proliferation [72]. Islets derived from type 2 diabetes patients showed reduced GRP94 levels [73]. Moreover, plasma of T1D patients contains elevated concentrations of free GPR94 as well as circulating GRP94-bound complexes such as GPR94- α1–antitrypsin [32] and GRP94-IgG [74,75]. The extracellular GRP94 represents an immunological danger as it stabilizes the bound peptide [76,77] and increases the efficiency of uptake by scavenger receptor type A (SR-A) and CD91 for cross-presentation [78,79,80,81]. It may therefore be of interest to explore the presence of GRP94 bound to its clientele, such as GRP94-PI complex, in circulation in T1D patients as a potential therapeutic marker.

Another study showed an increased macrophage polarization, accompanied by inflammatory environment and development of insulin resistance in conditional GRP94 KO in macrophages in mice [82], suggesting a connection between GRP94 and associated molecules in facilitating inflammatory response which should be tested also in β-cells with loss of function for GRP94. Another element that bridges ER stress to inflammation are viral infections, as cells are burdened with massive synthesis of viral proteins which may result in an increase of endogenous proteins misfolding [83] as well as induction of proinflammatory environment [84].

A third possible connection between mishandled proinsulin and inflammation may be UPR activation. Activation of classical UPR is known to promote inflammatory signaling [59]. PERK/eIF2α activation-mediated IκBα translational inhibition [53] or rapid proteasomal degradation [85] may decrease IκB/NF-κB ratio and facilitate nuclear translocation of NF-κB as discussed in the previous paragraph. So theoretically, GRP94 KO-induced ER stress and inflammation can intersect at multiple levels and may lead to inflammatory settings that trigger the immune response against β-cells as proposed in Figure 8.

## 5. Limitations of the Study/Further Possibilities

The role of proinsulin-derived peptides that are revealed as potential neoepitopes following GRP94 KO or cytokine exposure in the induction of autoimmunity needs to be investigated. A preliminary further analysis could have been made using bioinformatics tools such as NetMHC [86], which uses trained artificial neural networks to make predictions for strong MHC class-I binding. However, this tool is not trained/optimized for rat strain-based haplotypes due to literature scarcity. This is especially true for INS-1E cells, which carry RT1.A^g^ haplotype and for which no previous reports are available for comparison. In addition, the IEDB database lists only one MHC class-II epitope (FVKQHLCGSHLVEALYLV) derived from insulin in BB rats having RT1.A^u^ haplotype [87]. Furthermore, this approach of ER stress induction by limited β-cell folding capacity, leading to activation of inflammatory pathways, should be expanded to, for example, rodent models of diabetes such as NOD mouse/BB rats with tissue-specific GRP94 KO/targeted mutation or ER stress induced by thapsigargin in human cell lines/islets.

One more constraint in expanding our analysis is related to the less extensive proteome explored for rats (8133 reviewed proteins) compared to mice (17,085 reviewed entries) or humans (20,360 reviewed entries) on Uniprot until 20 January. Therefore, it is reasonable to deduce that we actually detected more RT1-bound peptides in our experimental settings but could not identify them in the database due to a relatively limited rat proteome data availability.

## 6. Conclusions

In conclusion, we found that β-cells challenged with a defect in proinsulin folding capacity displayed increased MHC I expression and an altered MHC I-bound immunopeptidome. This observation is consistent with the hypothesis that β-cell metabolisms and defects therein can result in the presentation of β-cell neoantigens that may contribute to the onset of diabetes. Here we show a clear link between the activation of inflammatory pathways in β-cells, which sensitizes them to immune attack as a possible sequalae to the onset of type 1 diabetes. Further experimental validation of our proposed idea in animal models and human cell lines are needed to advance this concept for translational benefits.

## Figures and Tables

**Figure 1 biomedicines-10-00814-f001:**
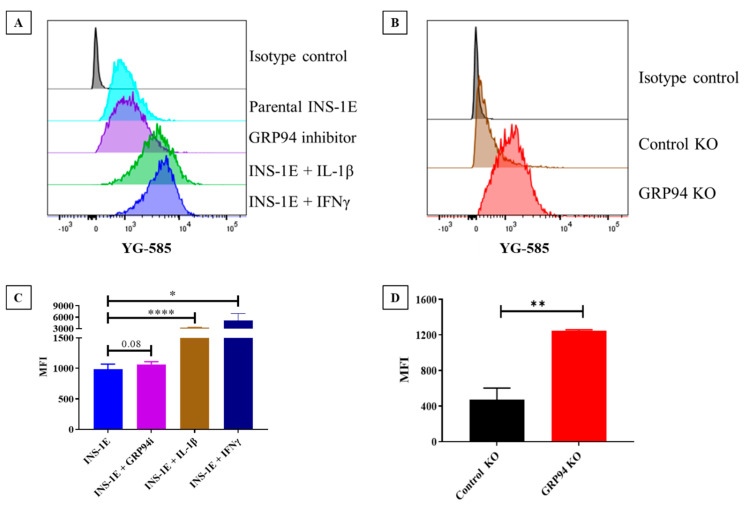
Flow cytometry comparison for RT1.A expression in INS-1E cells under tested conditions. (**A**,**B**) INS-1E cells compared with relevant controls for RT1.A expression and (**C**,**D**) statistical comparison for RT1.A expression INS-1E cells in tested conditions versus relevant controls. N = 3 for C and = 6 for D. MFI = mean fluorescence intensity. Drugs/cytokines dose and exposure: GRP94i = 20 µM, IL-1β = 15 pg/mL, IFNγ = 10 ng/mL for 24 h. * *p* ≤ 0.05, ** *p* ≤ 0.01, **** *p* ≤ 0.0001.

**Figure 2 biomedicines-10-00814-f002:**
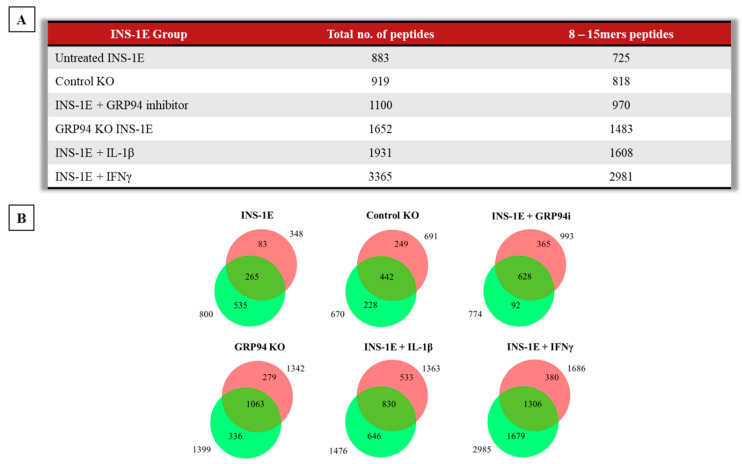
(**A**) Summary table of RT1.A-bound 8–15 amino acids length peptides from INS-1E cells under tested conditions. (**B**) Venn diagrams showing peptides overlap between the individual replicates per condition. *n* = 2 for all groups except for GRP94 KO where *n* = 4.

**Figure 3 biomedicines-10-00814-f003:**
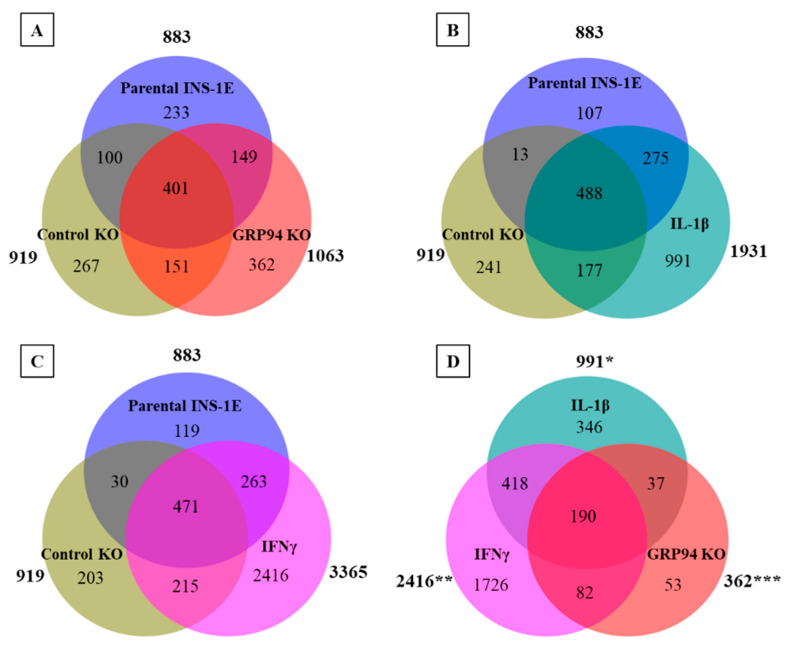
Venn diagrams showing number of MHC I-eluted peptides distinct or overlapping between (**A**) parental INS-1E, control KO & GRP94 KO, (**B**) parental INS-1E, control KO & IL-1β exposure (15 pg/mL for 24 h), (**C**) parental INS-1E, control KO & IFNγ exposure (10 ng/mL for 24 h), and (**D**) Peptides exclusive for IL-1β, IFNγ-exposed and GRP94 KO INS-1E groups. *n* = 2 for all except GRP94 KO where *n* = 4. * IL-1β peptides excluding parental INS-1E and control KO clone; ** IFNγ peptides excluding parental INS-1E and control KO clone; *** GRP94 KO peptides excluding parental INS-1E and control KO clone.

**Figure 4 biomedicines-10-00814-f004:**
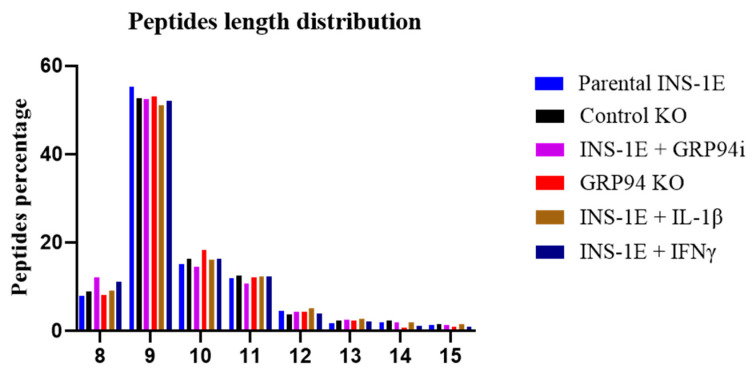
Length distribution for 8–15 amino acid-length peptides between parental INS-1E, control and confirmed GRP94 KO INS-1E, 24 h treatment with GRP94 inhibitor (20 µM), or INS-1E cells exposed to cytokines IL-1β (15 pg/mL) or IFNγ (10 ng/mL). *n* = 2 for all except GRP94 KO where *n* = 4.

**Figure 5 biomedicines-10-00814-f005:**
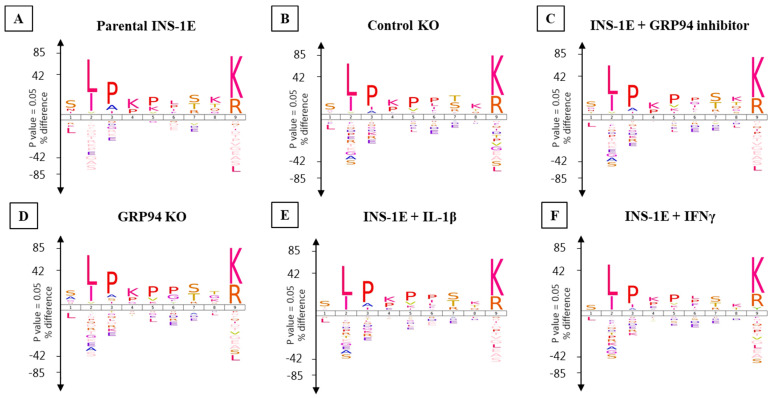
Nonamers binding motif for INS-1E cells under tested conditions. (**A**) Parental INS-1E, (**B**) control GRP94 KO clone, (**C**) INS-1E treated with GRP94 inhibitor (20 µM for 24 h), (**D**) GRP94 KO INS-1E clone, (**E**) IL-1β exposure (15 pg/mL for 24 h), and (**F**) IFNγ exposure (10 ng/mL for 24 h). *n* = 2 for all except GRP94 KO where *n* = 4.

**Figure 6 biomedicines-10-00814-f006:**
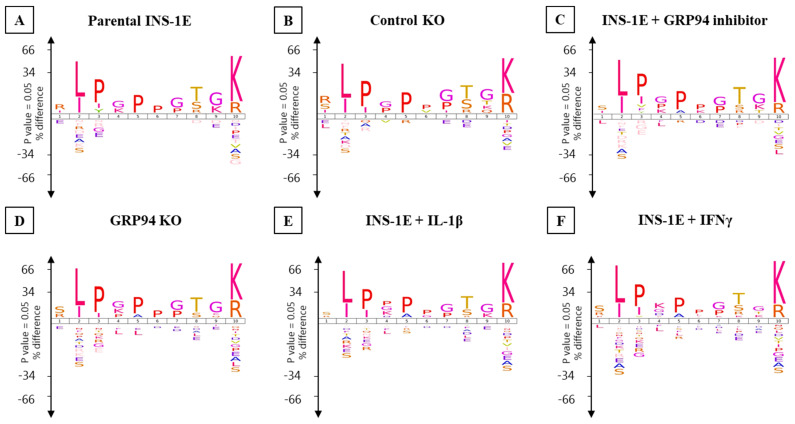
Decamers binding motif for INS-1E under tested conditions. (**A**) Parental INS-1E, (**B**) control GRP94 KO clone, (**C**) INS-1E treated with GRP94 inhibitor (20 µM for 24 h), (**D**) GRP94 KO INS-1E clone, (**E**) IL-1β exposure (15 pg/mL for 24 h), and (**F**) IFNγ exposure (10 ng/mL for 24 h). *N* = 2 for all except GRP94 KO where *N* = 4.

**Figure 7 biomedicines-10-00814-f007:**
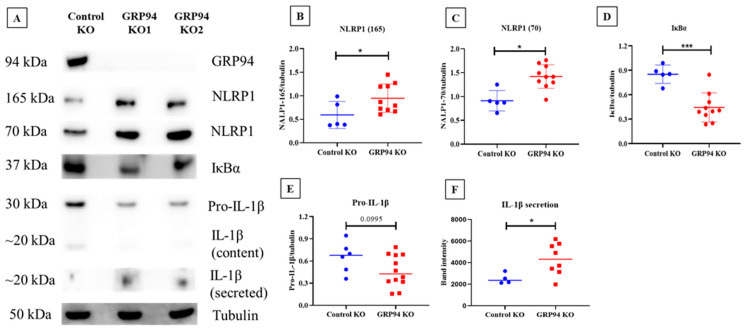
Western blot analysis for the expression of NLRP1, IκBα, and IL-1β in GRP94 KO INS-1E and control KO cells. (**A**) SDS-PAGE for expression of NLRP1 (fully assembled version (165 kDa) and without LRR (70 kDa)), IκBα, and IL-1β in tested conditions. (**B**–**F**) Statistical analysis (unpaired *t*-test) for the expression of NLRP1, IκBα, and pro-IL-1β/IL-1β, quantified and normalized to tubulin in tested conditions. *n* = 4–6 for different proteins of interest. (**B**–**E**) were generated using the ratio of quantified protein of interest to tubulin while (**F**) was generated based on quantified IL-1β only from the supernatant from equal cell numbers cultured and volume of culturing media. * *p* ≤ 0.05, *** = *p* ≤ 0.01.

**Figure 8 biomedicines-10-00814-f008:**
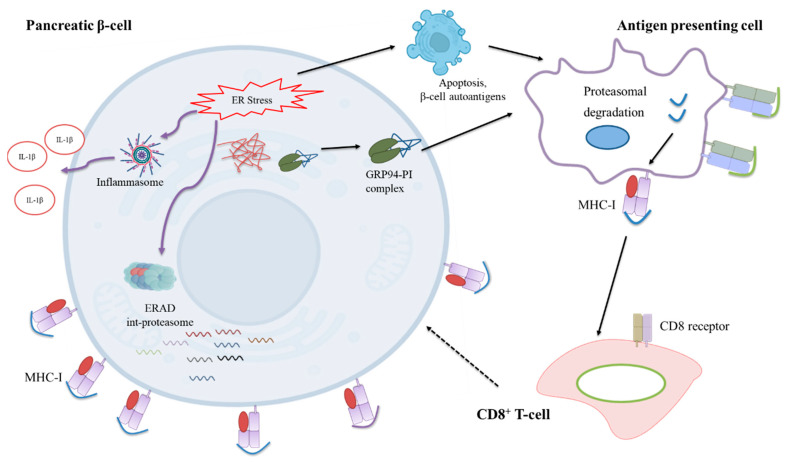
Proposed mechanism of autoimmunity induction in T1D. A defect in the ability of GRP94 to process proinsulin leads to an accumulation of mishandled proinsulin in β-cells. This leads to ER stress, activation of inflammatory pathways, assembly of int-proteasome, and secretion of GRP94-PI complexes. Simultaneously, the increased MHC-I expression on β-cells make them more visible to immune cells. The secreted GRP94-PI complexes are taken up by APCs and cross-presented to CD8^+^ T-cells to trigger immune attack against β-cells.

**Table 1 biomedicines-10-00814-t001:** RT1.A eluted proinsulin peptides from INS-1E cells derived from (pre)-proinsulin overlapping or unique for GRP94 KO/KD or cytokines exposure. ‘✓’refers to the presence of peptide in the group whereas empty column indicates absence of the peptides in the specific group. *n* = 2 for all groups except GRP94 KO where *n* = 4.

(Pre-Pro) Insulin Peptides	INS-1E		Control KO		INS-1E + GRP94i		GRP94 KO INS-1E				INS-1E + IL-1β		INS-1E + IFNγ	
	Rep 1	Rep 2	Rep 1	Rep 2	Rep 1	Rep 2	Rep 1	Rep 2	Rep 3	Rep 4	Rep 1	Rep 2	Rep 1	Rep 2
VLWEPKPAQAFVK											✓	✓	✓	✓
ILWEPKPAQAFVK													✓	✓
ALWMRFLPL													✓	✓
HLVEALYL							✓	✓	✓		✓		✓	
ALYLVCGERGFF					✓	✓	✓	✓	✓				✓	✓
ALYLVCGERGFFYTP					✓		✓	✓					✓	✓
YLVCGERGFF					✓		✓	✓			✓		✓	
FFYTPKS					✓		✓	✓	✓					
VEDPQVPQ					✓		✓	✓	✓					

## Data Availability

The raw data acquired by mass spectrometry will be deposited at ProteomeXchange Consortium via the PRIDE data repository. A request to access the raw data and results can be sent to anthony.purcell@monash.edu.

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
