# Peer review of "Defective Proinsulin Handling Modulates the MHC I Bound Peptidome and Activates the Inflammasome in β-Cells"

_biomedicines, 2022, doi:10.3390/biomedicines10040814_

Round 1

Reviewer 1 Report

I have found this manuscript interesting, timely, and comprehensive. The authors are specific to the relevant objectives of the study. All the assays are correctly described and informed.  I have a few suggestions for authors;

1. Authors must pay close attention to word spacing, font size, and significant values.

2. I suggest that significant limits should be included in the statistical analysis section.
3. Authors must elaborate conclusions in order to make them more understandable to the reader.
4. To valid your in vitro findings, I highly advise you to do in vivo experiments (animal models) with the same models.
5. Overall, the findings of the study are very clear.

Author Response

Biomedicines - 1611248

Khilji et al. 2022 - Defective proinsulin handling modulates the MHC I bound peptidome and activates the inflammasome in β-cells

Response to Reviewer comments

  1. Authors must pay close attention to word spacing, font size, and significant values.

The word spacing and font size have been adjusted at lines 37, 157, 309, 369, 468, 545, 549 and scheme 1. The font size has been adjusted for figures 1 – 4. The significant values are explained in the statistical analysis section (lines 235 – 237). Also, the individual significant values are explained in relevant figure legends (lines 268 & 400).

  1. I suggest that significant limits should be included in the statistical analysis section.

We have now added significant limits in the statistical analysis section (lines 235 – 236).

  1. Authors must elaborate conclusions in order to make them more understandable to the reader.

Conclusions are now been modified and made clearer according to our findings presented in the manuscript (lines 527 – 536).

  1. To validate your in vitro findings, I highly advise you to do in vivo experiments (animal models) with the same models.

It is an interesting and quite relevant suggestion. We have modified our conclusions section to reflect our intentions to advance this concept in animal models for translational benefits.

  1. Overall, the findings of the study are very clear.

We thank you for your appreciation of our study.

Reviewer 2 Report

Summary:

In the present paper, the authors present their research results that was carried out in order to demonstrate that the proinsulin mishandling, ER stress and the inadequate ER proinsulin folding capacity can sensitise β-cells to immune attack.

The hypothesis is very interesting and current, the study is original and the study design is appropriate. The authors presented their results in a logical and meaningful manner, using tables and figures for the better interpretation of the data. The conclusions of the work are clear and are well supported by the results.

The authors contribute to this field of research with new and valuable findings.

Author Response

In the present paper, the authors present their research results that was carried out in order to demonstrate that the proinsulin mishandling, ER stress and the inadequate ER proinsulin folding capacity can sensitise β-cells to immune attack.

The hypothesis is very interesting and current, the study is original and the study design is appropriate. The authors presented their results in a logical and meaningful manner, using tables and figures for the better interpretation of the data. The conclusions of the work are clear and are well supported by the results.

The authors contribute to this field of research with new and valuable findings.

We are glad for the appreciation of our work and acceptance for the special issue of the journal Biomedicines.

The hypothesis is indeed quite interesting and we will continue working on the same idea for translational relevance.